# Cutaneous Squamous Cell Carcinoma: From Biology to Therapy

**DOI:** 10.3390/ijms21082956

**Published:** 2020-04-22

**Authors:** Roberto Corchado-Cobos, Natalia García-Sancha, Rogelio González-Sarmiento, Jesús Pérez-Losada, Javier Cañueto

**Affiliations:** 1Instituto de Biología Molecular y Celular del Cáncer (IBMCC)-Centro de Investigación del cáncer (CIC)-CSIC, Laboratory 7, 37007 Salamanca, Spain; rober.corchado@usal.es (R.C.-C.); nataliagarciasancha@usal.es (N.G.-S.); jperezlosada@usal.es (J.P.-L.); 2Instituto de Investigación Biomédica de Salamanca (IBSAL), Complejo Asistencial Universitario de Salamanca, Hospital Virgen de la Vega, 37007 Salamanca, Spain; gonzalez@usal.es; 3Molecular Medicine Unit, Department of Medicine, University of Salamanca, 37007 Salamanca, Spain; 4Department of Dermatology, Complejo Asistencial Universitario de Salamanca, 37007 Salamanca, Spain

**Keywords:** cutaneous squamous cell carcinoma, immunotherapy, epidermal growth factor receptor (EGFR) inhibitors, BRAF inhibitors, azathioprine, cyclosporine

## Abstract

Cutaneous squamous cell carcinoma (CSCC) is the second most frequent cancer in humans and its incidence continues to rise. Although CSCC usually display a benign clinical behavior, it can be both locally invasive and metastatic. The signaling pathways involved in CSCC development have given rise to targetable molecules in recent decades. In addition, the high mutational burden and increased risk of CSCC in patients under immunosuppression were part of the rationale for developing the immunotherapy for CSCC that has changed the therapeutic landscape. This review focuses on the molecular basis of CSCC and the current biology-based approaches of targeted therapies and immune checkpoint inhibitors. Another purpose of this review is to explore the landscape of drugs that may induce or contribute to the development of CSCC. Beginning with the pathogenetic basis of these drug-induced CSCCs, we move on to consider potential therapeutic opportunities for overcoming this adverse effect.

## 1. Introduction

Cutaneous squamous cell carcinoma (CSCC) is the second most frequent cancer in humans, with an estimated incidence of 1 million cases each year in the US. This figure continues to rise, and is an underestimate [1,2]. The number of CSCCs has increased from 50% to 300% in the last three decades [3], and by 2030 its incidence in European countries will be twice the current level [4]. It is estimated that the risk of developing a CSCC at some point in life is 7% to 11% in the Caucasian population [5] (from 9% to 14% in men and from 4% to 9% in women) [6].

While it usually exhibits benign clinical behavior, it can be locally invasive and metastatic. Ten-year survival after surgery exceeds 90% for CSCC, but drops dramatically when metastases occur [7]. The frequency of lymph node metastases is around 4%, and mortality rates are nearly 2%. Given its high frequency, CSCC has a significant impact on overall mortality [8]. It is the second most common cause of death from skin cancer after melanoma and is responsible for the majority of deaths from skin cancer in people older than 85 years [3]. In some areas of the US, it has a mortality comparable to that of renal, oropharyngeal, and melanoma carcinomas [3]. 

CSCC arises from the malignant proliferation of epidermal keratinocytes. There are environmental and constitutional risk factors for its development. With respect to the former, older age, male sex, fair skin, immunosuppression, and a previous history of actinic keratosis (AK) are of known importance. Chronic sun exposure is the most important and well-known environmental factor associated with CSCC [9,10,11,12,13,14]. Solid-organ transplant recipients, who have a human papillomavirus infection or chronic lymphocytic leukemia, have a higher risk of developing CSCC than the general population [15,16,17,18]. AK is considered a premalignant lesion that may progress to an invasive CSCC, and is the most significant predictive factor of CSCC [19]. 

Several molecular pathways have been implicated in CSCC development. Ultraviolet-induced *P53* mutations are early events in CSCC, and are responsible for great genomic instability [10,20]. CSCC has the greatest mutational burden of all solid tumors, which, as we will see later, has therapeutic implications [21]. Other genetic changes occur subsequently in other suppressor genes, such as *CDKN2A* and *NOTCH* [22,23], and in oncogenes, such as *RAS* [24]. The accumulation of mutations ultimately involves various signaling pathways [25], including the activation of the NF-kB, MAPK, and PI3K/AKT/mTOR pathways [26,27], which mediate epidermal growth factor receptor (EGFR) overexpression. Epigenetic changes may also occur [28]. 

Surgery is the cornerstone of the management of CSCC, and radiotherapy is sometimes also implemented. However, a subset of patients with locally advanced and metastatic CSCC may benefit from systemic treatments [29]. The signaling pathways involved in CSCC development have given rise to targetable molecules in recent decades. Moreover, the high mutational burden and increased risk of CSCC in patients under immunosuppression were part of the rationale for developing the immunotherapy for CSCC that has changed the therapeutic landscape in recent years [30]. This review focuses on the molecular basis of CSCC and the current biology-based approaches of targeted therapies and immune checkpoint inhibitors. Another purpose of this review is to explore the landscape of drugs that may induce CSCC. Beginning with the pathogenetic basis of these drug-induced CSCCs, we move on to consider potential therapeutic opportunities for overcoming this adverse effect.

## 2. Molecular Basis of CSCC

Cutaneous squamous cell cancer is one of the most highly mutated human cancers [21,31]. A deeper knowledge of the molecular basis of CSCC would be useful for developing better ways of treating this disease.

The mutation of the tumor suppressor gene *TP53* has an important role early in the pathogenesis of CSCC and occurs in 54%–95% of cases [10,20,32]. Mutations of *TP53* are induced by ultraviolet radiation (UVR), the most important environmental risk factor for CSCC, and are reported in pre-malignant AK lesions and CSCC [33,34]. UVR-induced mutagenesis results in characteristic C-T and CC-TT dipyrimidine transitions, which enable tumor cells to prevent apoptosis and to promote clonal expansion of p53 mutant keratinocytes [35]. The role of *p53* in ultraviolet B-induced carcinogenesis has been confirmed in *p53*^−/−^ mice, which have an increased propensity for developing AK lesions and CSCCs secondary to ultraviolet B (UVB) exposure [36,37]. Furthermore, several groups have confirmed the presence of *p53* mutations in CSCC cell lines [38,39]. *P53* mutations are an early event in CSCC development and are ultimately responsible for great genomic instability. 

Other mutations subsequently occur in tumor suppressors, such as *CDKN2A* and *NOTCH*, and in oncogenes, such as *RAS*. [22]. The *CDKN2A* gene encodes two alternatively spliced proteins, p16INK4a and p14ARF. The inactivation of the *CDKN2A* locus may be due to loss of heterozygosity, point mutations, and promoter hypermethylation [23]. Loss of function of either p16INK4a or p14ARF may lead to unrestrained cell cycling and uncontrolled cell growth mediating pRB [40] and p53 [41]. On the other hand, loss of function *NOTCH1* and *NOTCH2* mutations are identified in more than 75% of CSCCs [42]. In vivo mouse studies show that *Notch1* deletion, a mutation that occurs early in CSCC, results in the development of skin tumors and facilitation of chemically-induced skin carcinogenesis [43,44]. The *Notch1* gene is a direct target of *p53* [45], and keratinocyte-specific ablation of *Notch1* disrupts the balance between growth and differentiation [46]. The upregulation of the Wnt/beta-catenin pathway, which may result from Notch1 loss of function, facilitates skin tumor development and promotion [43], and is at least partly dependent on p21WAP/Cip1 [47]. In vivo studies of *Notch1*-deficient mouse skin showed an increase in fibroplasia, angiogenesis, and inflammation, demonstrating the importance of the stromal microenvironment in CSCC development [48].

Loss of the *NOTCH1* gene may have cooperative effects with Ras-activation in keratinocyte transformation [22,45]. Regarding *RAS* genes, *HRAS* mutations (3%–20% of CSCCs), rather than *NRAS* and *KRAS,* are commonly associated with CSCC [21,31]. *Ras* has been implicated in the initiation of CSCC in a murine chemical carcinogenesis model [49], and mediating CDK4, in the induction of cell cycle arrest and transformation of primary keratinocytes into invasive carcinoma [50]. *HRAS* mutations were found at a higher frequency in CSCC lesions arising in melanoma patients treated with BRAF-inhibition [51]. RAS activation promotes upregulation of downstream MAPK and PI3K/AKT/mTOR intracellular signaling. These pathways, in non-*RAS* mutant CSCCs, may also result from alternative mechanisms, including EGFR overexpression or PTEN inactivation.

EGFR overexpression is common in CSCC, and is associated with the acquisition of a more aggressive phenotype and a poor prognosis [26,52]. EGFR is a member of the ErbB family of tyrosine kinase receptors that transmit a growth-inducing signal to cells that have been stimulated by an EGFR ligand. The union of ligand with EGFR produces a conformational change that allows a homodimerization with another EGFR or heterodimerization with another ErbB family member, both of which induce activation [53]. The pathways affected by the activation of EGFR include RAS-RAF-MEK-MAPK, PLC-gamma/PKC, and PI3K/AKT/mTOR. STAT and NF-kB can also be activated [54]. All these pathways are frequently altered in tumors, including CSCC [55], and trigger increased proliferation, migration, survival, resistance to apoptosis, and altered differentiation. The EGFR and downstream pathways can both be targeted with a variety of drugs to inhibit CSCC progression, as discussed below.

Therefore, epigenetic events play important roles in AK and CSCC [56]. CSCC includes the promoter hypermethylation of previous genes, such as *p16INK4a* and *p14ARF*, as well as *CDH1*, *RB1, MGMT*, and *RASSF1*, among others. These genes are involved in cycle regulation, DNA repair, epithelial adhesion, and signal transduction, while hypermethylation of CpG islands in the promoter regions produces transcriptional silencing [28]. MicroRNAs also have an important role; some act as oncogenes and others as tumor suppressors [57], and some are regulated by epigenetic factors. Recurrent copy number aberration has been noted in the development of CSCC (loss of heterozygosity at 3p, 8p, 9p, 9q, 13q, and 17q and chromosomal gain of 11q and 8q), including the formation of isochromosomes, chromosomal deletions, and whole-arm translocation [58]. 

Finally, the tumor microenvironment is important in the carcinogenesis of CSCC [59], attracting greater attention as its relevance in tumor development has become apparent [60,61]. One of the main components of the tumor microenvironment is inflammation [61], which may act as a tumor promoter [62,63]. The lack of inflammatory response is relevant in tumor progression [64]. Recent studies demonstrate that the CSCC tumor microenvironment is enriched in cancer-associated fibroblasts (CAFs) [65] and tumor-associated macrophages [66]. Tumor stromal cells are implicated in the invasion, metastases, tumor progression, and response to chemotherapy [67,68]. Cellular and molecular components of the tumor microenvironment are of great importance in the effect of immunotherapy, as described below. 

## 3. Treatment of CSCC

### 3.1. Targeted Therapy in CSCC

#### 3.1.1. EGFR Inhibitors

Current strategies in cancer therapy have pointed towards the interruption of signaling pathways that are involved in its pathogenesis. EGFR inhibitors were one of the first systemic therapies tested to treat CSCC. Some studies demonstrated that EGFR could be relevant to CSCC development, and in the context of the low effectiveness of drugs for treating CSCC, this was a logical and promising pathway to explore. EGFR inhibitors were tested in other cancers and yielded reasonable responses [69,70,71,72], and some isolated cases showed an anti-EGFR response in CSCC [73,74,75,76,77], prompting the design of clinical trials.

Targeting EGFR inhibits the PI3K/AKT/mTOR and RAS/RAF/ERK signal transduction pathways [78]. There are two classes of EGFR inhibitors: monoclonal antibodies that block the extracellular domain of the receptor (e.g., cetuximab, panitumumab, nimotuzumab, zalutumumab), and small-molecule tyrosine kinase inhibitors (TKIs), which block tyrosine kinase activity and thereby inactivate downstream cellular pathways (e.g., gefitinib, erlotinib, afatinib, lapatinib, neratinib, dacomitinib). Monoclonal antibodies and TKIs have been evaluated in clinical trials for poor-prognosis CSCC but are currently off label.

Cetuximab is a human-mouse chimeric monoclonal antibody that competitively binds to the extracellular domain of EGFR and inhibits dimerization of the receptor and the subsequent downstream signaling. Cetuximab is a U.S. Food and Drug Administration (FDA)-approved drug for colorectal and head and neck cancers and has shown some clinical efficacy as a first-line treatment in patients with unresectable CSCC [79]. Cetuximab was the first EGFR inhibitor to be evaluated in CSCC in a phase II trial. In that study, cetuximab showed valuable clinical activity with an overall disease control rate of 69% and a response rate (RR) of 28% at six weeks, including two complete remissions (6%) and eight partial remissions (22%). To confirm these results, a larger clinical trial (NCT03325738) is currently underway. Cetuximab is also being tested in combination with radiotherapy (NCT01979211), lenvatinib, which is a TKI (NCT03524326), avelumab, which is an anti-PD-L1 checkpoint inhibitor (NCT03944941), pembrolizumab, which is directed against as programmed cell death 1 protein (PD-1) (NCT03082534), and before surgery, as a neoadjuvant therapy (NCT02324608). Cetuximab is well-tolerated, but skin reactions may develop as side-effects in more than 80% of patients, mainly presenting as an acne-like rash, pruritus, desquamation, hypertrichosis, or nail disorders that must be treated [80,81,82]. The presence of acne-like eruption in patients under treatment has been associated with better response [79,83]. Another monoclonal antibody, panitumumab, was evaluated in 16 patients with incurable CSCC, five of whom (31%) showed a response [84]. Panitumumab is a good alternative to cetuximab when anaphylaxis occurs [85].

Small-molecule TKIs, like gefitinib, erlotinib, and lapatinib, have been partially effective in patients with CSCC. Gefitinib demonstrated modest activity in metastatic and locoregional recurrent CSCC with an overall RR of 16% and a disease control rate of 51% [86] (NCT00054691). Indeed, as neoadjuvant therapy before standard surgery or radiotherapy, gefitinib achieves a 45.5% RR in patients with aggressive or recurrent CSCC [87] (NCT00126555). In a single-arm phase II clinical trial, erlotinib exhibited a RR of 10% and progression-free survival (PFS) of 4.7 months in patients with recurrent or metastatic CSCC [88] (NCT01198028). Erlotinib has been used to inhibit EGFR in a three-dimensional in vitro human skin model, in which it resulted in a significant reduction of epidermal thickness [89]. Lapatinib, a dual TKI that blocks the HER2/neu and EGFR pathways, has been used to treat patients with CSCC and AK. It produced tumor reduction in two out of eight patients and AK reduction in seven out of eight patients, encouraging larger clinical trials [90]. In vitro studies demonstrate that lapatinib produces cell-cycle arrest, autophagy induction, and epithelial-to-mesenchymal inhibition in the CSCC A431 cell line [91].

The efficacy of EGFR inhibitors was somewhat lower than expected, and a better selection of patients should optimize the drug’s usefulness. It should be borne in mind that these targeted therapies, which inhibit signaling pathways that contribute to the CSCC progression, frequently disrupt skin homeostasis and produce side effects. 

#### 3.1.2. Other Targeted Therapies in CSCC

The involvement of RAS/RAF/MEK/ERK and PI3K/AKT/mTOR pathways in cancer has led to the development of several inhibitors that target them [92,93]. In CSCC, a recent in vivo study demonstrated that the inhibition of MEK with trametinib and cobimetinib induces senescence in CSCC cell lines and reduces tumor growth in a mouse model [94]. Moreover, cobimetinib is being studied in combination with atezolizumab, a PD-L1 inhibitor, in metastatic or locally advanced and unresectable CSCCs, and locally advanced CSCCs that are technically resectable but where surgery could produce disfigurement (NCT03108131). mTOR inhibitors such as rapamycin are currently being used to decrease the risk of CSCC development in immunosuppressed patients that receive traditional immunosuppression [95,96,97]. Combining topical mTOR inhibitors and AKT inhibitors (PHT-427) enhances the chemopreventive effects of rapamycin [98]. Pan-PI3K and selective PI3K inhibitors have been developed to treat other cancers [99]. In CSCC, GDC-0084 and LY3023414, which are novel small-molecule PI3K-mTOR dual inhibitors, inhibit survival and proliferation and promote apoptosis in CSCC cells. Moreover, these drugs inhibit A431 xenograft tumor growth [100,101]. Thus, targeting pathways downstream of EGFR could be a practical option for attacking CSCC. All the clinical trials that are currently being conducted with targeted therapies are listed in Table 1.

### 3.2. Immunotherapy in CSCC

Tumor cells produce neoantigens that are recognized and targeted by the immune system. When a T-cell recognizes the antigen expressed by the Human leukocyte antigen (HLA) complex in the tumor cell, co-receptors act as activators and inhibitors of the immune response [102]. Inhibitory receptors, such as programmed cell death 1 protein (PD-1) and Cytotoxic T-Lymphocyte Antigen 4 (CTLA4), are known as “immune checkpoint” receptors. PD-1 is an inhibitor co-receptor expressed on the surface of T-cells, B-cells, monocytes, natural killer cells, and dendritic cells [103]. This transmembrane protein binds to two ligands, PD-L1 and PD-L2, which are present on the surface of the tumor cell, and their interaction triggers a signal that inhibits the activated T-cells and induces immunological exhaustion via anergy and T-cell apoptosis [102,104,105]. The PD-L1/PD-1 axis is a primary mechanism of cancer immune evasion, and this was the rationale for developing new drugs that have emerged in recent years. Targeting the immune checkpoint proteins with monoclonal antibodies has yielded a clinical benefit in cancer [106,107], and dramatically changed prospects for the treatment of some types of cancer, such as melanoma [108]. An established tumor is composed both by the neoplastic cells and the tumor microenvironment. The latter is composed both by the tumor stroma and the inflammatory infiltrate. The tumor microenvironment, and not only the neoplastic cells, can also be modulated to destroy the neoplastic cells. Indeed, most immune checkpoint inhibitors are directed towards the lymphocytes, which belong to the tumor microenvironment, in order to enhance the immune response [109]. 

PD-1 inhibitors of several forms of cancer have been released, but given the low responsiveness of CSCC to other systemic treatments, some isolated cases were treated with drugs directed towards this axis and responded well [110,111]. These preliminary results justified closer examination of this pathway and its potential therapeutic role in CSCC. Some studies demonstrated the presence of cell surface PD-1/PD-L1 in human tumors, and this expression has been linked to poor clinical outcomes in a variety of cancers [112,113,114,115,116], including CSCC [117,118]. CSCC has the highest mutational burden of all tumors, and is a good candidate for immunotherapy treatment [21]. Tumors with a higher tumor mutational burden are known to be more responsive to immune checkpoint inhibitors [119,120,121]. In addition, the higher risk of immunocompromised patients developing CSCC indicates the importance of the immune system in this tumor [122,123]. For these reasons, clinical trials with these drugs for the treatment of CSCC were designed. 

Cemiplimab is the first drug approved by the FDA and the European Medicines Agency (EMA) for the treatment of locally advanced and metastatic CSCC [124]. It is a human monoclonal antibody directed against PD-1, and has demonstrated efficacy in immunocompetent patients with advanced CSCC and with metastatic disease, yielding RRs of 50% and 47%, respectively [124]. Cemiplimab is currently being tested in patients with recurrent stage III-IV head and neck CSCC before surgery as neoadjuvant therapy (NCT03565783), and in patients with recurrent CSCC as a pre-operative intralesional injection (NCT03889912). Future trials will focus on cemiplimab as an adjuvant drug versus placebo after surgery and radiotherapy in patients with high-risk CSCC (NCT03969004), as monotherapy, or in combination with RP1 oncolytic virus in patients with locally advanced or metastatic CSCC (NCT04050436). 

Other immunotherapeutic drugs are under evaluation in CSCC. Pembrolizumab is a human PD-1-blocking antibody indicated for the treatment of non-small-cell lung, head and neck, gastric, cervical, hepatocellular, and endometrial cancers, melanoma, Hodgkin’s lymphoma, and Merkel cell, urothelial, renal cell, small-cell lung, and esophageal carcinomas [125]. In CSCC, pembrolizumab is being tested in a phase II study of 150 adults with recurrent/metastatic or locally advanced unresectable CSCC (MK-3475-629/KEYNOTE-629, NCT03284424). The interim results of the preview clinical trial (CARSKIN, NCT02883556) presented at the American Society of Clinical Oncology (ASCO) meeting 2018 showed high RRs (42%) and a durable response, with a median of around seven months in patients with unresectable CSCC [126]. Pembrolizumab is also being examined in participants with locally advanced CSCC versus placebo after surgery and radiation (MK-3475-630/KEYNOTE-630, NCT03833167). It is being investigated as an addition to postoperative radiotherapy in resected cutaneous squamous cell cancer of the head and neck (NCT03057613) to assess safety with dose-limiting responses. Finally, pembrolizumab is being tested in combination with cetuximab (NCT03082534), AST-008 (NCT03684785), abexinostat (NCT035890054), and sonidegib (NCT04007744) in different stages of CSCC. 

Nivolumab, another PD-1 inhibitor, is being studied in patients with CSCC in monotherapy (NCT04204837, NCT03834233) or combination with pembrolizumab (NCT02955290), and there have already been case reports demonstrating its clinical efficacy and good tolerability [127]. Nivolumab is also being tested in combination with ipilimumab, an anti-CTLA-4 monoclonal antibody, in patients who are immunosuppressed due to having received a kidney transplant and who have unresectable or metastatic CSCC (NCT03816332). Pembrolizumab and nivolumab are FDA-approved for treating unresectable or metastatic melanoma but have yet to be approved for the treatment of CSCC. The most frequently reported side-effects of immune checkpoint inhibitors are diarrhea and fatigue, and they are usually low-grade side-effects. Immune checkpoint inhibitors can cause inflammation in any organ/system of the body, and thus it is important to take it seriously if the patient presents colitis, pneumonitis, hepatitis, thyroiditis, or hypophysitis. These autoimmune side-effects may sometimes be severe and force a treatment cycle to be discontinued or even withdrawn. Headache, pruritus, and dermatitis may be expected as well [128].

In addition to the evidence from clinical trials, there are several case reports of the efficacy of immunotherapy in CSCC-immunocompetent patients [129,130,131,132]. Transplant patients represent a group in which the use of checkpoint inhibitors presents a problem because enhanced T-cell activation can lead to allograft rejection [106,133,134]. Limited data exist because transplant patients are often excluded from clinical trials, and only data from isolated cases are available [130,135,136]. 

All the clinical trials with immunotherapy that are currently underway are listed in Table 2. Figure 1 shows the therapeutic landscape of CSCC. 

## 4. Pharmacologically Induced Cutaneous Squamous Cell Carcinoma

Several drugs have been developed for CSCC treatment, but the disease may actually be induced by drugs as well. Molecular mechanisms underlie pharmacologically-induced CSCC, and a sound knowledge of them could help physicians better tackle this tumor. Drug-induced CSCC is poorly covered in the literature, and for this reason, we focus on this CSCC in the last part of this review. 

### 4.1. Immunosuppressive Drugs and CSCC

The immunosuppressive therapy used in organ transplant recipients (OTRs) to prevent allograft rejection promotes cutaneous infection and skin neoplasms [15,122]. The classic immunosuppressant drugs used for organ transplantation are glucocorticosteroids (prednisone and prednisolone), calcineurin inhibitors (cyclosporine and tacrolimus), and anti-proliferative agents (azathioprine and mycophenolic acid). Here we focus on cyclosporine and azathioprine.

#### 4.1.1. Cyclosporine and CSCC

Cyclosporine is a calcineurin inhibitor that increases the risk of CSCC, especially under UVR [137,138,139]. Cyclosporine A reduces UVB-induced DNA damage repair and inhibits apoptosis in human keratinocytes by inhibiting the nuclear factor of activated T-cells (NFAT) [140]. Calcineurin inhibition is known to selectively induce the expression of activating transcription factor 3 (ATF3), which downregulated p53 expression and increased CSCC formation in a mouse model and in human CSCCs [141]. In vitro studies demonstrated that chronic treatment of human HaCaT keratinocytes with cyclosporine enhances AKT activation by suppressing PTEN, and promotes tumor growth of the CSCC A431 cell line in immune-deficient nude mice [142,143]. Furthermore, cyclosporine enhances epithelial-to-mesenchymal transition involving the upregulation of TGFβ signaling [144]. 

The increased risk of CSCC in patients under cyclosporine has led physicians to search for different options. Some studies of tacrolimus, a calcineurin inhibitor introduced to replace cyclosporine, demonstrated no difference in a comparison of overall cancer rates of the two drugs [145]; however, more recent data from a clinical trial and from in vivo studies indicate a lower skin cancer risk associated with tacrolimus [146,147]. Nevertheless, the most important drugs for preventing cyclosporine-induced CSCC development are the mTOR inhibitors.

The newest immunosuppressants used for OTRs are sirolimus (rapamycin) and everolimus. Both inhibit interleukin (IL)-2 and IL-15 via mTOR. It is not known whether these inhibitors have anticarcinogenic effects [148]. Preliminary data suggest that conversion from calcineurin inhibitors to sirolimus reduces the incidence of skin cancer in renal graft recipients [95,97], possibly because sirolimus reduces vascularization and the thickness of post-transplant CSCCs [149]. The change of therapy from calcineurin inhibitors to sirolimus in patients with one CSCC lowered the risk of a new CSCC, and metastasis events only occurred in patients who received calcineurin inhibitors [96], the effect being maintained over five years of follow-up [150]. In vivo studies of hairless mice show that sirolimus significantly increases the latency of large tumors and reduces their multiplicity. Tumors from the rapamycin group have a lower UV-signature p53 mutation rate [151]. Case reports of conversion to everolimus show a reduced likelihood of CSCC development [152].

Recent studies have shown that cyclosporine exposure upregulates IL-22R1 [153] and causes increased JAK1, STAT1, and STAT3 expression. Using ruxolitinib, an FDA-approved JAK1/2 inhibitor, in human CSCC cells and xenografts reduces proliferation and growth. This could be a feasible option for preventing CSCC in OTRs who face long-term immunosuppression [154]. 

#### 4.1.2. Azathioprine and CSCC

In a cohort study of 361 renal transplant recipients, the immunosuppressant drug azathioprine increased the risk of CSCC 2.4-fold [155]; and in an organ transplantation cohort of 207 patients, post-transplant azathioprine treatment increased the risk of CSCC compared with controls in a dose-dependent manner [156]. A systematic review and meta-analysis of 27 studies confirmed the association of OTRs treated with azathioprine and CSCC [157]. It is clear that azathioprine enhances the effect of UVR on skin cancer risk, and indeed, it strongly induces and promotes CSCC in hairless mice exposed to UVR [158]. Azathioprine photosensitizes the skin to UVR by changing the absorption interval of DNA upon incorporation of 6-thioguanine, the active metabolite of azathioprine. UVR absorption then induces the formation of reactive oxygen species that have been linked to DNA damage and cutaneous malignancies [159,160,161]. Whole-exome sequencing has revealed a novel CSCC mutational signature, which is associated with chronic exposure to azathioprine [39]. 

To reduce the risk of CSCC associated with this drug, azathioprine can be replaced by mycophenolate, leading to lower levels of DNA 6-thioguanine, skin ultraviolet A (UVA) sensitivity, and DNA damage, and a lower risk of CSCC [146,162,163]. However, another study suggests that the calcineurin inhibitor tacrolimus and mycophenolate mofetil (MMF) inhibit UVB-induced DNA damage repair, demonstrating the tumor-promoting action of these immunosuppressants [164].

#### 4.1.3. Voriconazole and CSCC

Voriconazole, an antifungal used to prevent and treat invasive fungal infections after lung transplantation, has been associated with an increased risk of developing CSCC [165]. Voriconazole causes photosensitivity [166] in a dose-dependent manner [167]. The mechanism underlying this may arise from a primary metabolite, voriconazole N-oxide, which absorbs UVA and UVB wavelengths [166,168]. Expression arrays of in vitro cultures of primary human keratinocytes exposed to voriconazole also show that this drug inhibits terminal epithelial differentiation pathways, resulting in poor formation of epithelial layers that are important for photoprotection, favoring its phototoxicity [169]. In vitro and in vivo assays demonstrated that voriconazole and its product inhibit catalase, raising intracellular levels of UV-associated oxidative stress and DNA damage in keratinocytes to promote skin carcinogenesis [170]. While photoprotection is fundamental for preventing CSCC, this is especially important in patients under voriconazole. 

### 4.2. Targeted Therapies

#### 4.2.1. Sonic-Hedgehog Inhibitors and CSCC

Medications to treat other skin cancers, such as melanoma and basal cell carcinoma (BCC), can paradoxically lead to the development of CSCC. Vismodegib is a smoothened inhibitor (Hedgehog pathway inhibitor) that the FDA and EMA have approved for treating locally advanced and metastatic BCC [171]. The association of vismodegib with CSCC was reported in several case reports [172,173,174], and a retrospective cohort study highlighted this increased risk [175]. Some researchers disputed the latter study [176], and a subsequent paper failed to replicate such an association [177]. Furthermore, squamous metaplasia has been found in BCCs treated with vismodegib [178]. Nevertheless, there is some evidence to suggest that hedgehog inhibitors may indeed increase the risk of CSCC. The mechanism of action of vismodegib to promote CSCC is thought to be the activation of the RAS/MAPK pathway, which is responsible for CSCC progression [179]. 

A CSCC may arise from a BCC because both develop from the same target cell, as some authors have suggested. Two studies revealed new roles for *Ptch1* that lie at the nexus between BCC and CSCC formation [180,181]. *Ptch1* gene is thought to occupy a critical role in determining the basal or squamous cell lineage [181], and its polymorphisms are involved in cell fate decisions. In BCC, loss of *Ptch1* activates the Sonic-Hedgehog pathway, but the overexpression of *Ptch1* promotes an alternative cell-fate decision, leading to increased CSCC susceptibility [180]. 

#### 4.2.2. BRAF Inhibitors and CSCC

*BRAF* is mutated in around 50% of melanomas, and some years ago, the therapeutic landscape of this tumor broadened through the development of BRAF inhibitors [182], specifically vemurafenib and dabrafenib [183]. These drugs provided greater overall survival and PFS compared with dacarbazine [184,185], but they also increased the risk of CSCC development [186,187,188]. The effectiveness of these drugs stems from their ability to attenuate the MAPK pathway, which is downstream of constitutive BRAF activation [189]. However, BRAF inhibitors are capable, paradoxically, of activating the MAPK pathway in cells containing non-mutated *BRAF*, and this pathway is essential for CSCC development [51,190,191,192]. The inhibition of MEK proved to be effective in preventing CSCC while on BRAF inhibitors, and thereafter BRAF inhibitors were combined with MEK inhibitors to avoid these side effects. Specifically, vemurafenib is combined with cobimetinib [193], and dabrafenib with trametinib [194]. A meta-analysis of five phase III randomized controlled trials, 17 phase II trials, and two phase IV trials [195] demonstrated that combined BRAF and MEK inhibition (trametinib) reduces the incidence of CSCC relative to BRAF monotherapy, as seen in another study [196]. More recent work demonstrated that BRAF inhibitors induce *RAS* mutations that are essential for MAPK activation. *RAS* mutations were detected in 21%–60% of lesions after BRAF inhibitor treatment in contrast to 3%–30% in normal CSCCs [51,197]. A mutational signature has been noted in squamous proliferative lesions induced by BRAF inhibitors that differs from the mutation pattern seen in spontaneous CSCCs [198]. Additionally, human papillomaviruses (HPVs) are detected more frequently in BRAF inhibitor-induced CSCCs, which means that HPV might accelerate keratinocyte oncogenesis in this subset of patients [199].

Other than MEK inhibitors, the inhibition of cyclooxygenase (COX)-2 has been evaluated as a strategy to prevent BRAF-inhibitor-mediated CSCC development. Experimental investigations that induce CSCC carcinogenesis by UVR have shown that COX-2 inhibitors (celecoxib and diclofenac) decrease prostaglandin production, thereby mitigating CSCC development [200,201]. Moreover, celecoxib delayed the onset of CSCC in a mouse model mediated by DMBA/TPA and of CSCC induced by the BRAF inhibitor PLX7420, reducing the tumor burden by 90% [202]. All the drugs that may contribute to the development of CSCC are listed in Table 3.

## 5. Conclusions

In recent years, a deeper understanding of the molecular bases of cutaneous squamous cell carcinogenesis (CSCC) has helped identify novel therapies. EGFR inhibitors were found to be promising drugs in CSCC, based on several studies that suggested an important role for this pathway in CSCC development at a time when there was little to offer patients by way of effective treatment. Subsequently, other targets were evaluated and continue to be developed. More recently, the high mutational burden of this tumor and the increased risk of CSCC in immunosuppressed patients have raised the possibility of using immunotherapy to treat CSCC. As the new checkpoint inhibitors are surprisingly effective in other tumors, some CSCC cases have also been treated, with anti-PD-1 yielding particularly good responses. This prompted the design of clinical trials, and cemiplimab was the first inhibitor to be approved for use. It seems likely that other checkpoint inhibitors will be incorporated into the therapeutic arsenal of CSCC in the near future. 

It is important to emphasize that patients who are receiving drug treatments that are associated with increased susceptibility to developing CSCC may require dermatological supervision, especially if any suspicious skin lesion arises.

The major message emerging from our review is that we should guard against the view that CSCC is a tumor with a good prognosis simply because it usually has a favorable evolution. In truth, its high incidence means that the absolute frequency of complicated and disseminated cases will also be high.

Metastatic CSCC remains a therapeutic challenge. The new arsenal of drugs that target different signaling pathways, especially immunotherapeutic medications, opens up new possibilities for treating CSCC patients, and we may expect these to be increasingly incorporated into the new wave of personalized and precision medicine protocols.

## Figures and Tables

**Figure 1 ijms-21-02956-f001:**
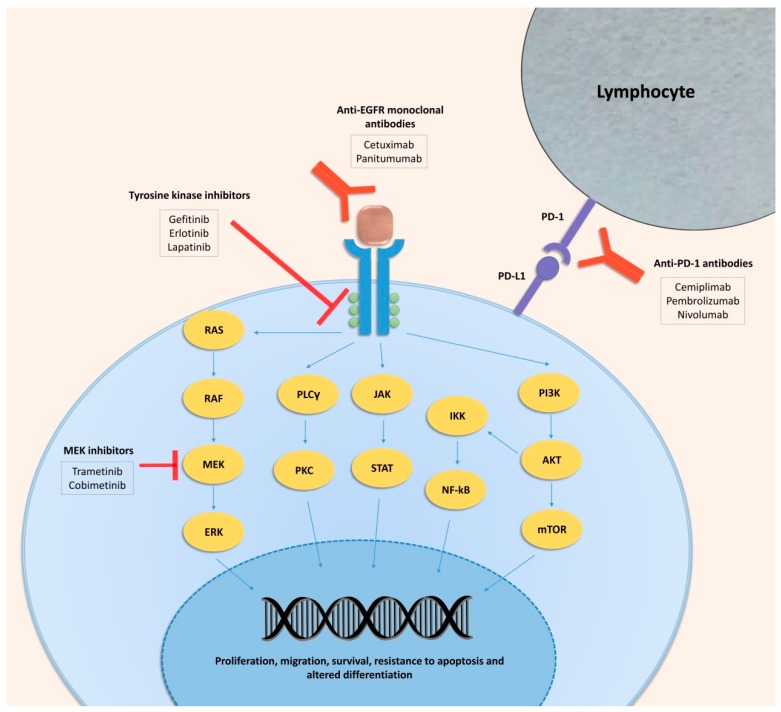
Therapeutic landscape of cutaneous squamous cell carcinoma.

**Table 1 ijms-21-02956-t001:** Clinical trials of targeted therapies in cutaneous squamous cell carcinoma (CSCC) (revised until 29 January 2020).

Drug	Treatment	Conditions	Current State	NCT Code
**Cetuximab**	Alone	Locally advanced and metastatic CSCC surgically unresectable	Completed (28% response rate, 6% complete remission, 2% partial remission)	NCT00240682
	Alone	Locally advanced and metastatic CSCC surgically unresectable	Completed	NCT03325738
	Alone (neoadjuvant therapy)	Aggressive locally advanced CSCC	Recruiting	NCT02324608
	Combination with post-operative radiation	Locally advanced head and neck CSCC	Active, not recruiting	NCT01979211
	Combination with pembrolizumab	Recurrent/metastatic CSCC	Recruiting	NCT03082534
	Combination with lenvatinib	Advanced CSCC	Recruiting	NCT03524326
	Combination with avelumab	Advanced CSCC	Recruiting	NCT03944941
**Gefitinib**	Alone (neoadjuvant therapy)	Locally advanced/recurrent CSCC	Completed (45.5% response rate)	NCT00126555
	Alone	Metastatic or locorregional recurrent	Completed (16% response rate)	NCT00054691
**Erlotinib**	Alone	Recurrent/metastatic CSCC	Completed (10% response rate)	NCT01198028
	Combination with radiotherapy	Advanced head and neck CSCC	Completed	NCT00369512
	Alone (before surgery)	Head and neck CSCC	Active, not recruiting	NCT00954226
**Cobimetinib**	Combination with atezolizumab	CSCC	Recruiting	NCT03108131

**Table 2 ijms-21-02956-t002:** Clinical trials of immunotherapy in cutaneous squamous cell carcinoma (revised until 29 January 2020).

Drug	Treatment	Conditions	Current State	NCT Code
**Cemiplimab**	Alone	Advanced and metastatic CSCC	Completed (47%–50% response rate) Recruiting next phase	NCT02383212 NCT02760498
	Alone (before surgery)	Recurrent stage III-IV head and neck CSCC	Recruiting	NCT03565783
	Alone (pre-operative therapy intralesional)	Recurrent CSCC	Recruiting	NCT03889912
	Adjuvant therapy after surgery and radiotherapy	High risk CSCC	Recruiting	NCT03969004
	Alone or combination with RP1	Advanced or metastatic CSCC	Recruiting	NCT04050436
	Alone	Unresectable locally recurrent and/or metastatic CSCC	Recruiting	NCT04242173
	Alone (neoadjuvant therapy)	Stage II to IV CSCC	Recruiting	NCT04154943
**Pembrolizumab**	Alone	Recurrent/metastatic or locally advanced unresectable CSCC	Active, not recruiting	NCT03284424
	Alone	Locally advanced or metastatic CSCC	Active, not recruiting (preview results presented in ASCO show 42% response rate)	NCT02883556
	Alone	Locally advanced and metastatic CSCC	Active, not recruiting	NCT02964559
	Adjuvant therapy after surgery and radiotherapy	High risk locally advanced CSCC	Recruiting	NCT03833167
	Combination with postoperative radiotherapy	CSCC of head and neck	Recruiting	NCT03057613
	Combination with cetuximab	Recurrent/metastatic CSCC	Recruiting	NCT03082534
	Combination with AST-008	Advanced/metastatic CSCC	Recruiting	NCT03684785
	Combination with abexinostat	Stage III-IV CSCC of head and neck	Recruiting	NCT03590054
	Combination with sonidegib	Stage IV CSCC of head and neck	Not yet recruiting	NCT04007744
	Combination with nivolumab and CIMAvax vaccine	Stage III-IV CSCC of head and neck	Recruiting	NCT02955290
	Combination with SO-C101	Advanced/metastatic CSCC	Recruiting	NCT04234113
**Nivolumab**	Alone	Locally advanced/metastatic CSCC	Recruiting	NCT04204837
	Alone	Advanced CSCC	Recruiting	NCT03834233
	Alone or combination with ipilimumab	Metastatic CSCC in immunosuppressed patients	Recruiting	NCT03816332
	Combination with pembrolizumab and CIMAvax vaccine	Stage III-IV CSCC of head and neck	Recruiting	NCT02955290

**Table 3 ijms-21-02956-t003:** Pharmacologically-induced CSCC.

Drug	Treatment	Mechanisms to Promote CSCC	Options to Reduce CSCC Risk
Cyclosporine	Immunosuppressant	Reduces UVB-induced DNA damage repair and inhibits apoptosis by inhibiting nuclear factor of activated T-cells (NFAT) [140]	Sirolimus and everolimus [95,96,97,149,150,151,152]
		Induces the expression of ATF3, which downregulates p53 and increases CSCC formation [141]	
		Enhances AKT activation by suppressing PTEN and promotes tumor growth [142,143]	
		Enhances epithelial-to-mesenchymal transition involving the upregulation of TGFβ signaling [144]	
Azathioprine	Immunosuppressant	Photosensitizes the skin to ultraviolet radiation (UVR) by changing the absorption interval of DNA upon incorporation of 6-thioguanine and induces the formation of reactive oxygen species [159,160,161]	Mycophenolate mofetil [146,162,163]
Voriconazole	Antifungal	The primary metabolite, voriconazole N-oxide, absorbs UVA and UVB wavelengths and causes photosensitivity [166,167,168]	Photoprotection
		Inhibits terminal epithelial differentiation pathways resulting in poor formation of epithelial layers that are important for photoprotection [169]	
		Inhibits catalase, raising intracellular levels of UV-associated oxidative stress and DNA damage [170]	
Vismodegib (Sonic-hedgehog inhibitor)	Basal cell carcinoma	Activates RAS-MAPK pathway [179]	Close follow-up
Vemurafenib and dabrafenib (BRAF inhibitors)	Melanoma	Activate, paradoxically, MAPK pathway and induce *RAS* mutations [51,190,191,192,197]	BRAF inhibitors + MEK inhibitors [193,194,195,196] or BRAF inhibitors + cyclooxygenase (COX)-2 inhibitors [200,202]

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
