# Peer review of "Cutaneous Squamous Cell Carcinoma: From Biology to Therapy"

_ijms, 2020, doi:10.3390/ijms21082956_

Round 1
Reviewer 1 Report
CSCC is the one of the most frequent cancer in humans, with an estimated incidence of about a million cases each year in the US. Corchado-Cobos et al. nicely reviewed the molecular basis of CSCC and the current biology-based approaches of targeted therapies and immune checkpoint inhibitors. This is a timely and well thought review also focusing to explore the landscape of drugs that may induce or contribute to the development of CSCC. Authors need to provide responses and explanation to below comments.
Comments:
- How the toxicities will be overcome when the molecular targeted drugs are administered systemically as compared to topical application of the drugs that may have lower side-effects and unwanted toxicities.
- A table showing pharmacologically induced CSCC would be helpful.
Author Response
Reviewers' comments
Reviewer #1: Thank you for the edits. This is much improved and clearer. A few small suggestions:
-- How the toxicities will be overcome when the molecular targeted drugs are administered systemically as compared to topical application of the drugs that may have lower side-effects and unwanted toxicities.
ANSWER: We were not able to properly understand this suggestion. In our review, we have only focused on the current biology-based approaches of systemic targeted therapies and systemic immune checkpoint inhibitors. We have not discussed on topical medications.
We have commented on the most relevant side-effects of the different discussed treatments.
In page 4 line 168: “Cetuximab is well-tolerated, but skin reactions may develop as side-effects in more than 80% of patients, mainly presenting as an acne-like rash, pruritus, desquamation, hypertrichosis or nail disorders that must be treated [80-82]. The presence of acne-like eruption in patients under treatment has been associated with better response [79, 83]”.
In page 7, line 275 we have added a paragraph on the most common side-effects of immune checkpoint inhibitors, which we have missed in the previous version: “The most frequently reported side-effects of immune checkpoint inhibitors are diarrhea and fatigue, and they are usually low-grade side-effects. Immune checkpoint inhibitors can cause inflammation in any organ/system of the body, and thus it is important to take it seriously if the patient presents colitis, pneumonitis, hepatitis, thyroiditis or hypophysitis. These autoimmune side-effects may sometimes be severe and force to discontinue a treatment cycle or even withdrawn it. Headache, pruritus and dermatitis may be expected as well [128]”.
The particular management of the side-effects depend on the type and its severity and it is beyond the scope or this review.
-- A table showing pharmacologically induced CSCC would be helpful.
ANSWER: Many thanks for your suggestion. We have included the information of pharmacologically induced CSCC within the Table 3 (page 13)
Reviewer 2 Report
This manuscript, review article type, written by Dr. Roberto Corchado-Cobos et al., with title " Cutaneous Squamous Cell Carcinoma: From Biology to Therapy" focuses on the disease of cutaneous squamous cell carcinoma (CSCC).
Cutaneous squamous cell carcinoma is a malignant tumor arising from epidermal keratinocytes. In fair-skinned individuals, it typically develops in areas of photodamaged skin and presents with a wide variety of cutaneous lesions, including papules, plaques, or nodules, that can be smooth, hyperkeratotic, or ulcerated. The neoplasia can be of low-risk or high-risk (aggressive). The treatment for a low-risk lesion is usually a standard excision or other non-surgical destructive treatment (e.g. curettage, cryotherapy, etc…). The treatment for a high-risk lesion includes surgery, radiation, chemotherapy including immunomodulatory drugs such as anti-PD-1 and PD-L1.
In this review, the authors describe the molecular basis of CSCC and then make a thorough description of the current targeted therapies including EGFR inhibitors, immunotherapy, etc. This is followed by a revision of the pharmacologically induced CSCC, and a conclusion.
The manuscript is well written, it is easy to read and there are enough information and references. The tables and the figure are adequate. As a recommendation for improvement, I would suggest pointing out that some of the targeted therapies based on immunotherapy are targeting the tumor immune microenvironment. Therefore, are not targeting the neoplastic cell. If the authors could provide a histological figure showing the expression of PD-1, PD-L1, CTLA, etc…, the manuscript would be improved. Nevertheless, this is a final decision of the authors.
Author Response
Reviewer #2: Many thanks for your comments. We found their appreciation very interesting.
-- As a recommendation for improvement, I would suggest pointing out that some of the targeted therapies based on immunotherapy are targeting the tumor immune microenvironment. Therefore, are not targeting the neoplastic cell.
ANSWER: Many thanks for your recommendation. We have added in page 7 line 226 a paragraph pointing out this appreciation: “An established tumor is composed both by the neoplastic cells and the tumor microenvironment. The latter is composed both by the tumor stroma and the inflammatory infiltrate. Tumor microenvironment can be modulated to destroy the neoplastic cells. Indeed, most immune checkpoint inhibitors are directed towards the lymphocytes, which belong to the tumor microenvironment, in order to enhance the immune response”
-- If the authors could provide a histological figure showing the expression of PD-1, PD-L1, CTLA, etc…, the manuscript would be improved.
ANSWER: Unfortunately, we cannot provide such an image. These are not routinely performed staining in clinical practice to indicate the systemic treatment in CSCC and currently we are not using them in out department.